# UHMWPE/CaSiO_3_ Nanocomposite: Mechanical and Tribological Properties

**DOI:** 10.3390/polym13040570

**Published:** 2021-02-14

**Authors:** Sakhayana N. Danilova, Sofia B. Yarusova, Yuri N. Kulchin, Ivan G. Zhevtun, Igor Yu. Buravlev, Aitalina A. Okhlopkova, Pavel S. Gordienko, Evgeniy P. Subbotin

**Affiliations:** 1North-Eastern Federal University, 677000 Yakutsk, Russia; dbsksnsdjyj@mail.ru (S.N.D.); okhlopkova@ya.ru (A.A.O.); 2Institute of Chemistry, Far Eastern Branch, Russian Academy of Sciences, 690022 Vladivostok, Russia; yarusova_10@mail.ru (S.B.Y.); buravlev.i@gmail.com (I.Y.B.); pavel.gordienko@mail.ru (P.S.G.); 3Vladivostok State University of Economics and Service, 690014 Vladivostok, Russia; 4Institute of Automation and Control Processes, Far Eastern Branch, Russian Academy of Sciences, 690041 Vladivostok, Russia; kulchin@iacp.dvo.ru (Y.N.K.); s.e.p@list.ru (E.P.S.); 5Far Eastern Federal University, 8, Sukhanova St., 690091 Vladivostok, Russia

**Keywords:** polymer composite material, ultra-high-molecular weight polyethylene (UHMWPE), wollastonite (CaSiO_3_)

## Abstract

This paper studied the effect of additives of 0.5–20 wt.% synthetic CaSiO_3_ wollastonite on the thermodynamic, mechanical, and tribological characteristics and structure of polymer composite materials (PCM) based on ultra-high-molecular weight polyethylene (UHMWPE). Using thermogravimetric analysis, X-ray fluorescence, scanning electron microscope, and laser light diffraction methods, it was shown that autoclave synthesis in the multicomponent system CaSO_4_·2H_2_O–SiO_2_·nH_2_O–KOH–H_2_O allows one to obtain neeindle-shaped nanosized CaSiO_3_ particles. It was shown that synthetic wollastonite is an effective filler of UHMWPE, which can significantly increase the deformation-strength and tribological characteristics of PCM. The active participation of wollastonite in tribochemical reactions occurring during friction of PCM by infrared spectroscopy was detected: new peaks related to oxygen-containing functional groups (hydroxyl and carbonyl) appeared. The developed UHMWPE/CaSiO_3_ materials have high wear resistance and can be used as triboengineering materials.

## 1. Introduction

The issues of creating, studying, and using polymer composite materials (PCMs) are a promising and rapidly developing field of modern materials science. PCM are widely used for various purposes including the manufacture of parts of the most critical friction units operating in conditions of limited lubrication [1]. Due to the relatively low cohesive energy of polymers compared to metals and ceramics, PCM are able to effectively transfer/distribute the load across the polymer matrix and filler, which leads to a significant reduction in wear and friction [2,3]. The ability of PCM to dampen shock and vibration loads, along with excellent corrosion resistance, justifies and actualizes their use in the aerospace, chemical, transport, and marine industries. The ability to eliminate the use of lubricants to avoid pollution problems makes PCM parts in demand in application areas such as the textile and food industries.

To achieve the desired tribological/mechanical properties and other functions, various types of fillers are introduced into the polymer matrix (fibrous reinforcing fillers, finely divided fillers, chopped glass fiber, metal oxides, etc.) [4,5,6,7,8,9,10,11,12,13,14,15,16,17]. Matrix fillers have an advantage over a mixture of fillers such as fibers, since they do not require alignment and orientation in composites, which significantly affects the wear resistance of PCM and directed loading when slipping [18]. 

Among the most common polymer matrices for PCM is ultra-high-molecular weight polyethylene (UHMWPE). UHMWPE, being a linear polyolefin with a high molecular weight (2–10.5) × 10^6^ g/mol, has good mechanical and tribological properties [19]. Combined with resistance to various aggressive chemical environments and a low coefficient of friction, UHMWPE is a universal material for use in harsh operating conditions including low temperature [20,21]. Composite materials based on UHMWPE, filled with various micro- and nanoscale fillers, have been developed for use as structural and functional materials in industry, which significantly expands the scope of application [22,23,24].

A number of studies have shown the effectiveness of using wollastonite (calcium monosilicate Ca_6_Si_6_O_18_ with a theoretical composition, wt.%: CaO—48.3, SiO_2_—51.7 [25,26,27,28,29]) to increase the strength and durability of PCM [30,31,32,33,34]. Wollastonite is used in various high-tech fields including the ceramics and polymer industry [35,36,37,38]. The use of wollastonite as a filler in PCM is associated with its reinforcing effect on the polymer matrix, chemical inertness, high melting point, and environmental friendliness [38]. The introduction of wollastonite in polymeric materials is primarily aimed at improving the mechanical properties of PCM such as elastic modulus, yield strength, and elongation at break, which have a strong effect on the resistance to mechanical deformation [39]. A high ratio of wollastonite grain length to diameter provides better load stress transfer from the polymer matrix to the filler when exposed to an external load, which leads to an increase in the mechanical properties of PCMs [38,40,41,42]. PCMs containing wollastonite are used in many industries, covering a wide range of products: from thermoset and thermoplastic materials and products to automotive and aerospace products. It is assumed that wollastonite has a reinforcing effect on the polymeric matrix of UHMWPE, which causes the load transfer to the reinforcing component. In the long-term, PCM developed based on UHMWPE with a wollastonite filler is possible for use as a material for friction units of machinery (bearings, plain bearings, sliding bearings, piston rings). Thus, wear studies are an unavoidable element in this study, as the durability and reliability of friction components are crucial in the optimal operation of the entire technical system.

The feedstock for the synthesis of wollastonite is a variety of compounds of calcium and silicon. Currently, a number of methods for producing wollastonite have been developed, among which one can distinguish melt production methods; synthesis by direct solid-state reactions; and synthesis at lower temperatures, based on the interaction of reacting components in aqueous media. The latter includes hydrothermal (autoclave) and hydrochemical synthesis followed by dehydration of the resulting calcium hydrosilicates to produce wollastonite. These groups of methods are currently most actively studied, since they are promising from the standpoint of minimizing energy costs [43,44,45,46]. The establishment of the influence of the type and ratio of the initial components of the reaction mixture and synthesis conditions on the composition, structure, morphology, and functional properties of wollastonite as well as the systematization of the results for various multicomponent systems remains relevant at the present time. It should be noted that the components of the CaSO_4_·2H_2_O–SiO_2_·nH_2_O–KOH–H_2_O model reaction system in which needle wollastonite was obtained can be replaced by calcium- and silicon-containing wastes of the corresponding composition, which is planned to be shown by the authors in further studies.

Thus, the aim of this paper was to study the effect of synthetic needle wollastonite obtained hydrothermally in the multicomponent system CaSO_4_·2H_2_O–SiO_2_·nH_2_O–KOH–H_2_O on the physical, mechanical, and tribological characteristics and structure of PCM based on UHMWPE.

## 2. Materials and Methods

### 2.1. Synthesis of Wollastonite

Wollastonite was produced in the multicomponent system CaSO_4_·2H_2_O–SiO_2_·nH_2_O–KOH–H_2_O under autoclave treatment of the corresponding reaction mixture at a temperature of 220 °C for 3 h. The initial components (calcium sulfate, 2-aqueous CaSO_4_·2H_2_O, grade p.a., silicon dioxide, grade pure, potassium hydroxide, grade p.a.) were mixed in a stoichiometric ratio. After a specified time interval, the precipitate was separated from the solution by filtration through a blue ribbon paper filter, washed with distilled water heated to 60–70 °C, and dried at 85 °C for 5 h. The degree of reaction flow was monitored by the residual potassium hydroxide concentration in the solution. To obtain wollastonite CaSiO_3_, the residues after synthesis were calcined in the temperature range of 900–1000 °C for 1 h.

### 2.2. Obtaining Polymer Composite Materials (PCMs)

To obtain PCMs, UHMWPE grade GUR-4022 (Celanese, Nanjing, China), with a molecular weight of 5.3·10^6^ g/mol, an average particle size of 145 μm and a density of 0.93 g/cm^3^, was used as a polymer matrix. The PCM components were mixed in a paddle mixer at a rotor speed of 1200 rpm. The concentration of wollastonite in UHMWPE was 0.5, 1, 2, 5, 10, and 20 wt.% Samples for research were obtained by hot pressing at a temperature of 175 °C and a pressure of 10 MPa holding for 20 min, followed by cooling to room temperature.

### 2.3. Methods of Analysis

X-ray diffraction patterns of the samples were recorded on the D8 ADVANCE automatic diffractometer (Bruker, Karlsruhe, Germany) with sample rotation in Cu K_α_ radiation. X-ray phase analysis (XRF) was performed using EVA search software with the XRF database Powder Diffraction FileTM (Soorya N Kabekkodu, 2007). Card numbers: 00-027-0088—wollastonite CaSiO_3_ of monoclinic modification; 01-084-0654—wollastonite CaSiO_3_ of triclinic modification; 00-033-0306—calcium hydrosilicate Ca_1.5_SiO_3.5_·xH_2_O/1.5CaO·SiO_2_·xH_2_O; 01-089-6459—tobermorite 9 Å Ca_4_(Si_6_O_15_)(OH)_2_·5H_2_O.

The particle size distribution of the powders was found using a laser particle analyzer Analysette-22 NanoTec/MicroTec/XT (Fritsch, Germany). Measurements were taken three times for each sample, then the results were averaged.

The specific surface was identified by the method of low-temperature nitrogen adsorption using the Sorbtomer-M device (Novosibirsk, Russia). The thermal behavior of the sample was studied using the synchronous thermal analysis device NETZSCH STA 449 °C Jupiter^®^ (Selb, Germany) in the temperature range 20–1000 °C.

The morphology of the microparticles of the obtained samples was studied using a scanning electron microscope (SEM) with field emission SIGMA 300 VP (Carl Zeiss, Oberkochen, Germany).

The mechanical properties of UHMWPE and PCM were studied using the Autograph AGS-J tensile testing machine (Shimadzu, Tokyo, Japan). The tensile strength and elongation at break were tested according to ASTM D3039/D3039M-14 at the moving gripper speed of 50 mm/min, the number of samples was six.

The tribological characteristics of UHMWPE and PCM were determined according to ASTM G99–17 using a universal testing machine UMT-3 (CETR, Mountain View, CA, USA). The pin-on-disk testing method was used for tribological characterization. The scheme of “pin-on-disk” friction is shown in Figure 1. Samples had a diameter of 10.00 ± 0.02 mm and a height of 20.0 ± 0.2 mm. The counterbody was a disk made of tempered unalloyed carbon steel AISI 1045 with a hardness of 56–58 HRC and a roughness of R_a_ = 0.06–0.08 μm. The specific load was 1.9 MPa, and the linear slipping velocity was 0.5 m/s. Test time of each sample was 3 h.

The wear rate (mm^3^/N⋅m) was estimated as follows:(1)$$\mathrm{Wear}\;\mathrm{rate}=\frac{V_{\mathrm{lost}}}{F_{N}\cdot d}=\frac{m_{\mathrm{lost}}}{\rho \cdot F_{N}\cdot d}$$
where V_lost_ is the volume lost during sliding (mm^3^); m_lost_ is the mass lost during sliding (g); ρ is the density of the composite; F_N_ is the normal force (N); and d is the sliding distance (m). To calculate the wear rate, we measured the mass of the UHMWPE and PCM samples before and after friction on Discovery DV215CD analytical scales (OHAUS, Greifensee, Switzerland) with an accuracy of 0.00001 g.

The hardness (H) of the samples was measured by pressing the ball ISO 2039/1-87 on the UMT-3 universal testing tribometer. The diameter of the steel ball indenter was 5.0 ± 0.05 mm, accuracy of indentation depth measurement was ±0.005 mm, the indentation time ±0.1 s, and the load was 132 N. The sample was 30 × 30 mm in size and 4.0 ± 0.1 mm thick.

The supramolecular structure and friction surface of UHMWPE and PCM were studied on the JSM-7800F scanning electron microscope (Jeol, Akishima, Japan) with the X-MAX-20 attachment (Oxford Instruments plc, Tubney Woods, Abingdon, UK) in the secondary electron mode at an accelerating voltage of 1–1.5 kV.

Fourier transform infrared (IR) spectroscopy (FTIR; Varian 7000, Palo Alto, CA, USA) was used to record IR spectra with an attenuated total reflection (ATR) attachment over the range 400–4000 cm^−1^. In the study of changes in the surface layer of PCM, the IR spectra were obtained before and after friction tests.

The thermodynamic characteristics of UHMWPE and composites were studied on a DSC 204 F1 Phoenix NETZSCH differential scanning calorimeter (Selb, Germany), where the measurement error was not more than ±0.1%, the heating rate was 20 °C/min, and the sample weight was 18 ± 1 mg. The measurements were carried out in a helium medium in a temperature range of 40–180 °C. The samples were placed in aluminum crucibles with a 40 µL. Temperature calibration was performed using standard samples of In, Sn, Bi, Pb, and KNO_3_. The degree of crystallinity of UHMWPE and PCM was calculated by Equation (2):(2)$$\alpha ,\;\%=\frac{\Delta H_{endothermal}}{\Delta H_{f}\left(1-W_{f}\right)}\cdot 100\%$$
where $\Delta H_{endothermal}$ is the melting enthalpy calculated from the area of endothermic melting peak; $\Delta H_{f}$ is the melting enthalpy for 100% crystalline UHMWPE, which is equal to 291 J/g; and *W_f_* is the mass content of the filler in PCM [19,47].

## 3. Results and Discussion

### 3.1. Synthesis of Wollastonite

The autoclave synthesis product in the CaSO_4_·2H_2_O–SiO_2_·nH_2_O–KOH–H_2_O system after drying the sample at 85 °C had a specific surface area of 146.4 m^2^/g and is characterized by the presence of an amorphous phase, monoclinic wollastonite (a = 15.42600; b = 7.32000; c = 7.06600; α = 90.000; β = 95.400; γ = 90.000.), calcium hydrosilicate Ca_1.5_SiO_3.5_·xH_2_O/1.5CaO·SiO_2_·xH_2_O, and tobermorite 9 Å Ca_4_(Si_6_O_15_)(OH)2·5H_2_O (a = 6.73500; b = 7.38500; c = 22.48700; α = 90.000; β = 90.000; γ = 123.250). A thermogravimetric analysis (TG) (Figure 2) showed that the synthesized sample contained 17.6% of the water released in the temperature range from 20 to 700 °C. At 820 °C, an exoeffect was recorded on the thermogram, which relates to the transition of amorphous hydrated forms of calcium silicates to the crystalline phase of wollastonite, which was confirmed by the XRF data on the non-volatile residue upon calcination to the indicated temperatures. After calcination at a temperature of 900 °C for 1 h, the phase composition of the sample was characterized by the presence of a phase of triclinic modification wollastonite CaSiO_3_ (a = 7.92580; b = 7.32020; c = 7.06530; α = 90.055; β = 95.217; γ = 103.426) (Figure 3). It can be seen from the infrared spectrum (Figure 4) that the wollastonite obtained was characterized by a group of absorption bands in the region of 850–1100 cm^−1^, associated with asymmetric vibrations of Si–O–Si bridge bonds as well as with asymmetric and symmetric vibrations of Si–O terminal bonds. The group of bands in the region of 550–750 cm^−1^ was assigned to symmetric vibrations of Si–O–Si bridge bonds in the [SiO_4_] tetrahedra. Absorption bands in the low-frequency region 400–550 cm^−1^ can be associated with deformation vibrations of the O–Si–O terminal bonds and vibrations of calcium-oxygen bonds in the [CaO_6_] octahedra [48,49,50].

After calcination at 900 °C, the specific surface area of the CaSiO_3_ sample was 26.4 m^2^/g.

An analysis of the particle size distribution, confirmed by SEM images (Figure 5), indicates that the obtained CaSiO_3_ sample consisted of agglomerates with sizes from 1 to 70 μm. Agglomerates consisted of needle-shaped CaSiO_3_ particles less than 100 nm in diameter. It should be noted that the histogram (Figure 5) does not show the distribution of a high content of free nanoparticles in the sample volume, which indicates a PCM hardening mechanism due to the reinforcement of the polymer with micron-sized agglomerates (including the spherulitic type).

### 3.2. Description of Properties of PCM

#### 3.2.1. Morphology of PCM

To assess the distribution of filler particles in the polymer and explain the results of the mechanical characteristics, structural studies were performed by the SEM method (Figure 6 and Figure 7). The structure of the original UHMWPE was characterized as lamellar (Figure 6a). With the introduction of wollastonite (Figure 6c) into PCM, the formation of spherulites of the radial type with structural elements of different sizes was observed. Wollastonite is randomly distributed in the polymer matrix (Figure 6b–d) and contains particles with a needle-like structure represented by fibers with a diameter of about 35 nm and a length of 100–200 nm (Figure 7). With an increase in the content of wollastonite, the presence of finely dispersed particles 1~50 μm in size with a well-developed surface structure was recorded.

#### 3.2.2. Mechanical Properties of PCM

When filling UHMWPE with 2 wt.% of wollastonite, in PCM, there was an increase in the value of elongation at break and tensile strength (Table 1). The introduction of filler into the polymer matrix led to an increase in tensile strength by 27% compared to unfilled UHMWPE, and to a slight increase in elongation at break by 14%. Further addition of the filler led to a gradual decrease in these characteristics. The highest increase in the Young’s modulus was observed with the 10 wt.% wollastonite addition to UHMWPE, with an 81% increase in the Young’s modulus index relative to the initial polymer (Table 1). Consequently, the stiffness of the polymer composition increased, preventing the development of deformation during tensile strain of the material, while the elasticity of the material became worse.

The hardness of the composites increased as the content of wollastonite in UHMWPE increased. The addition of 20 wt.% wollastonite increased of hardness value by 42% is observed. The increase in the hardness of the polymer composition with the addition of the filler was due to the strengthening of the polymer itself due to a certain orientation of the macromolecules.

Based on the SEM studies (Figure 5) of the autoclaved synthesis product heat-treated at 900 °C in the CaSO4·2H_2_O–SiO_2_·nH2O–KOH–H2O system, triclinic wollastonite has a nanoscale fibrous structure, which ensures its activity as a modifying agent during crystallization of UHMWPE with a transformation of the supramolecular lamellar structure of UHMWPE into spherulite, also providing additional polymer hardening due to mechanical adhesion of the wollastonite needle particles to the polymer. In this case, due to load transfer from the matrix to the fiber, hardening of the composite material was observed with an increase in the deformation characteristics of the polymer [51]. The PCM’s spherulite structure increases material stiffness and load bearing capacity [52,53,54,55]. Reducing the size of the structural components of the composite will require an increase in the energy spent to destroy such a system, which explains the increase in strength and modulus of elongation of PCM [6].

Thus, the formation of needle-shaped crystals with a high anisotropy factor determines the efficiency of using wollastonite as a hardening component of PCM.

#### 3.2.3. Thermodynamic Properties of PCM

To assess the effect of wollastonite on the properties and structure of UHMWPE, thermodynamic studies were carried out using the differential scanning calorimetry (DSC) method (Table 2). Most PCMs are made taking into account the preservation of one of the main parameters: heat resistance. Any decrease in the heat resistance of PCM with the introduction of fillers is not desirable.

The introduction of wollastonite into the polymer matrix in an amount of 0.5, 1.0, 2.0, 5.0, 10, and 20 wt.% does not lead to a change in the melting point of PCM; this parameter remains within the measurement error (127.6–128.1 °C). With increasing filler content in UHMWPE, the enthalpy of melting gradually decreased compared to the starting polymer. In the case of a composite containing 20 wt.% of wollastonite, there was a decrease in the enthalpy of melting by ≈11% relative to UHMWPE. A decrease in the melting enthalpy with an increase in the amount of introduced CaSiO_3_ indicated an increase in the melt viscosity of the composites and a limitation of the mobility of polymer macromolecules as a result of interaction with the filler surface [56]. In addition, agglomerated filler particles act as a barrier to the crystallization of UHMWPE.

It was found that the addition of 10 and 20 wt.% wollastonite leads to an increase in the crystallinity degree by 4 and 10% relative to the initial UHMWPE, respectively. The increase in the degree of crystallinity at high wollastonite content may indicate an increase in the size of UHMWPE crystallites. It can be assumed that the model wollastonite with a regular needle-like structure contributes to structural modification with an improvement of the composite supramolecular structure. It is known that as the degree of crystallinity increases, the density, stiffness, and hardness of PCM increase, while the elasticity and flexibility of the material deteriorate (Table 1). The dependence of the strength of materials on the degree of crystallinity was shown earlier [57]. It was assumed that the Young’s modulus and hardness of the material increases with increasing degree of crystallinity. This confirms and explains the increase in the Young’s modulus and hardness of PCM containing 10 and 20 wt.% wollastonite.

#### 3.2.4. Tribological Properties of PCM

Studies have shown that when filling UHMWPE with 1 wt.% wollastonite, a decrease in linear wear by two times and wear rate by six times is observed, while the coefficient of friction remains within the measurement error (Table 3). Consequently, the composites are characterized by optimal tribological characteristics. However, the lowest indicators of linear wear were observed in the composite containing 10 wt.% wollastonite, where a decrease in the linear wear rate by 2.8 times was noted. The increase in wear resistance of PCM filled with wollastonite is possibly due to the fact that nanofiller particles reduce the contact area of PCM with the metal surface of the counterbody, along with orientation effects, with the arrangement of the surface layers of the composite in the slipping direction (Figure 8).

However, when exceeding 2 wt.% of wollastonite content, deterioration of wear resistance was observed (i.e., the wear rate indicator increases). This is probably connected with the fact that some part of the agglomerated particles of wollastonite during friction migrates to the friction surface of the PCM and to the counterbody surface. Formed solid particles can act as abrasive particles in the friction path of the material, which accelerates the increase in the wear rate. It was also noted that in all composites, the friction coefficient was maintained at the level of the original polymer, regardless of the filler content.

In general, the tribological characteristics of polymers are determined from two aspects of the tribosystem: the contact conditions and the microstructure of the polymer samples, which are sensitive to the temperature in the friction zone [1]. It is known [58] that the friction of UHMWPE causes the solid ledges on the rough surface of the steel counterbody to deform the polymer surface with its subsequent displacement. As a result of this, on microphotographs of wear surface of PCM, the formation of furrows oriented along the friction direction was observed (Figure 8). Thus, the wear particles accumulated on a friction surface, interacting with particles of a filler (wollastonite), forming the focused intermediate film. The formed friction film had a higher modulus of elasticity than the original polymer due to its higher density, but the UHMWPE/CaSiO_3_-based PCM exhibited a mixed wear pattern upon friction. It is known that at high pressure and slipping speed, filler particles are removed from the matrix, causing an increase in the degree of wear. Since the granulometric composition of wollastonite is characterized by large (up to 70 μm), although in the polymer volume it reaches up to ~50 µm, agglomerates that are heterogeneous in shape and size, it is very difficult to achieve convergence of results with increasing concentration of the introduced filler. Detachment of fillers and their presence in the friction zone leads to abrasion of the polymer material and its subsequent removal in the form of wear particles as friction contact continues. As shown in Table 2, with an increase in wollastonite content above 2 wt%, there was an increase in wear rate. The effect of broken fillers and their abrasive effects on the material was reported [59] as the dominant mechanism of PCM failure during friction. Large particles of agglomerate filler during friction carry extreme loads. Stress transfer and their concentration at the matrix–filler interface can also lead to significant deformations of UHMWPE. Due to the repeated high stresses and deformations, specific zones lose their bearing capacity, and filler breaks out from the PCM. The resulting layer of products of mechanical destruction of wollastonite has an abrasive effect in tribocontact, in contrast to the lubricating effect. Therefore, the improvement of the mechanical and tribological characteristics of PCM containing CaSiO_3_ has certain limits by the criteria for its concentration in UHMWPE.

#### 3.2.5. Investigation of the Friction Surface of PCM

##### Wear Surface Morphology

To explain the increase in the wear resistance of PCM, the structure of the friction surfaces of UHMWPE and composites was investigated by the method of scanning electron microscopy (SEM). On the SEM images (Figure 8), it was registered that the increase in wollastonite content led to a change in the morphology of the friction surface of PCM.

SEM images (Figure 8) showed that an increase in the content of wollastonite led to a change in the morphology of the friction surface of PCM. The friction surface of all composites was characterized by grooves oriented in the direction of friction, which was the result of plastic deformation of the polymer in contact with the solid surface of the steel counterbody. In PCM containing 1 wt.%, (Figure 8b) of wollastonite, the formation of an inhomogeneous structure of the friction surface without the above grooves was established, while an increase in the wear resistance by six times in comparison with the initial UHMWPE was recorded. With an increase in the content of wollastonite in PCM, the depth of the friction grooves decreased, and an oriented secondary structure was formed as a result of the accumulation of wear products and filler particles. Due to the formation of a shielding structure, an increase in the wear resistance of the material due to the localization of shear deformations during friction was observed. In addition, agglomerates of wollastonite particles and wear products protrude onto the slipping surface, thereby absorbing the main load [60].

##### Fourier Transform Infrared (FTIR)

Figure 9 presents the results of studies of the PCM surface before and after friction by IR spectroscopy. The absorption bands of 2915 cm^−1^, 2847 cm^−1^, 1472 cm^−1^, and 719 cm^−1^, which are related to the absorption bands of the initial UHMWPE, were recorded on the IR spectra of PCM before friction. The IR spectra before PCM friction, containing 2, 5, 10, and 20 wt.% of wollastonite, had absorption bands in the range 850–1100 cm^−1^, which correspond to asymmetric vibrations of Si–O–Si bridge bonds as well as asymmetric and symmetric vibrations of Si–O terminal bonds.

As can be seen from Figure 9b, in the IR spectra after friction, the appearance of new peaks in the region of 3000–3600 cm^−1^, related to the associated hydroxyl groups –OH, and in the region of 1560–1700 cm^−1^, corresponding to vibrations of carbonyl groups –C=O, was recorded. The observed phenomenon indicates the occurrence of oxidative processes in the area of frictional contact. After friction of the composites, the peak intensity of the absorption bands of Si–O–Si vibrations increase, which indicates the localization of wollastonite particles on the friction surfaces of the material. It has been recorded that an increase in the content of wollastonite in PCM is accompanied by an intensification of oxidative processes.

The introduction of CaSiO_3_ nanoparticles into the UHMWPE polymer matrix also improves the tribological properties of PCM. During PCM wear, complex tribochemical reactions occur, of the removal of material in the form of wear particles from the tribocontact zone, which depends on the friction conditions, the structure of the polymer chain, the presence of functional groups in the polymer, surface roughness, and the behavior of polymer chains in the process of tribooxidation. Following from the results of IR spectroscopic studies of the friction surface of PCM samples after tribological tests (Figure 9), the PCM friction process based on UHMWPE/CaSiO_3_ is accompanied by tribochemical reactions with a change in the intensity of oxidation processes in the frictional contact zone. It should be noted that the amount of CaSiO_3_ filler introduced into the polymer matrix affects the change in the intensity of tribooxidation. It was shown that an increase in the content of wollastonite enhances the processes of tribooxidation, while the less thermostable UHMWPE can be removed from the friction zone in the form of wear particles, and the more heat-resistant wollastonite is gradually localized on the friction surfaces (Figure 8). Thus, the restructuring of the PCM structure under the influence of shear deformations was established, due to which a new, more wear-resistant, oriented structure was formed (Figure 8). On this surface, heat-resistant wollastonite particles are elements that protrude on the surface and absorb the load, which dramatically reduces the actual contact area.

High concentrations of wollastonite in UHMWPE lead to the fact that the fillers no longer serve as the crystallization centers of the polymer, a result of which the PCM structure becomes loose and disordered. For samples prior to tribological testing, differences in the IR spectra of PCM (bands for the Si–O–Si groups) were observed for wollastonite contents from 2 to 20 wt.%. For samples after friction tests, a difference was observed in the intensity of IR spectra for the Si–O–Si groups at a wollastonite content of 1 to 2 wt.%.

Thus, it can be established that the studied wollastonite contributes to the increase in the deformation-strength and tribotechnical characteristics of PCM even at low filler contents.

## 4. Conclusions

This paper studied the effect of synthetic wollastonite additives with a specific surface area of 26.4 m^2^/g, obtained by autoclave synthesis at 220 °C in a model multicomponent system CaSO_4_·2H_2_O–SiO_2_·*n*H_2_O–KOH–H_2_O system on the physical, mechanical, and tribological characteristics and structure of PCM based on UHMWPE. It is established that the synthesized wollastonite is an effective modifier of UHMWPE, enabling an increase in the deformation-strength characteristics and increased the tribological parameters of the composite. The optimum concentrations of wollastonite in UHMWPE, corresponding to 1.0–20 wt.% were determined. Therefore, the greatest improvement in indicators of wear resistance was observed in the composite containing 1 wt.% wollastonite, where the wear rate decreased by six times. Through the mechanical property indices, the composite containing 2 wt.% wollastonite was characterized by an increase in the tensile strength value by 27% and elongation at break by 14%. It is shown that such a change in the material properties is associated with the transformation of the UHMWPE supramolecular structure from lamellar to spherulitic. It has been recorded that wollastonite particles, despite their agglomeration, are the centers of UHMWPE crystallization and form spherolites with distinct boundaries. Their sizes depend on the filler content: an increase in wollastonite concentration in UHMWPE up to 10 and 20 wt.% resulted in an increase in the degree of crystallinity of PCM, the structure of which was identified as loosened, accompanied by a decrease in the elasticity of the material. However, these composites had high indices of hardness and Young’s modulus: hardness increased by 42% and Young’s modulus by 81% at 10 wt.% filler content. It was found that wollastonite is also active in the processes of friction and wear of PCM. Through IR spectroscopy, the occurrence of tribooxidative reactions upon the introduction of wollastonite in UHMWPE was recorded: new peaks appear to be related to oxygen-containing functional groups (hydroxy- and oxo- groups) and their intensity increased with increasing filler content. The structure of the friction surfaces of PCM was characterized as homogeneous, oriented in the direction of slipping, with localization of wollastonite particles on the surface. Thus, the restructuring of the PCM structure under the influence of shear deformations and the formation of a secondary, more wear-resistant surface on which wollastonite particles are the load-bearing elements, which sharply reduces the actual contact area, was established.

A new tribotechnical material UHMWPE/CaSiO_3_ was developed, characterized by increased deformation and strength characteristics, and wear resistance for friction units of machinery and mechanisms.

## Figures and Tables

**Figure 1 polymers-13-00570-f001:**
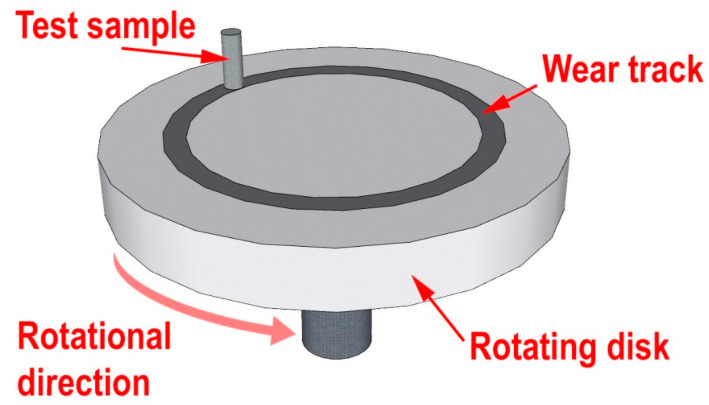
Schematic configuration of the tribology machine employed for the wear test.

**Figure 2 polymers-13-00570-f002:**
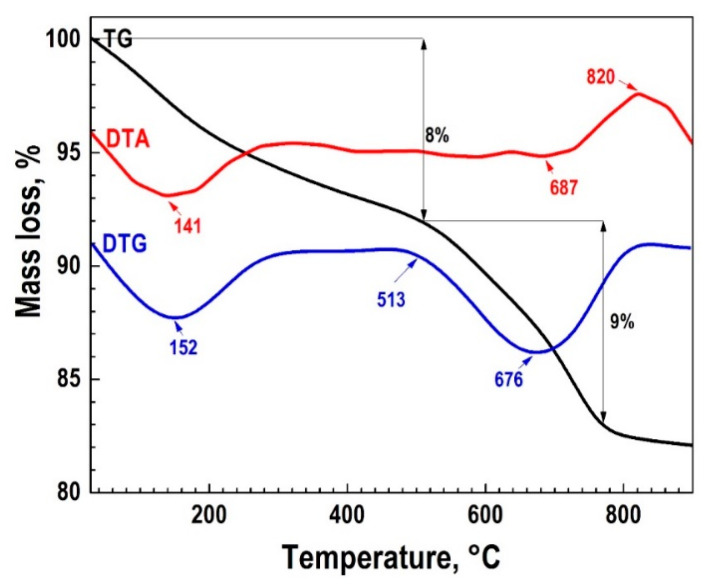
Thermogravimetric analysis (TG) of the autoclave synthesis reaction product at 220 °C in the CaSO_4_·2H_2_O–SiO_2_·*n*H_2_O–KOH–H_2_O system.

**Figure 3 polymers-13-00570-f003:**
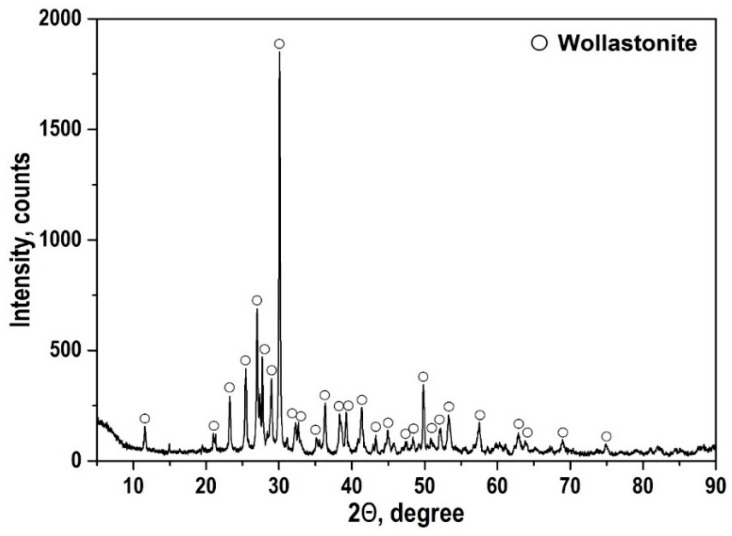
Phase composition of the autoclave synthesis product at 220 °C in the multicomponent system CaSO_4_·2H_2_O–SiO_2_·*n*H_2_O–KOH–H_2_O after calcination at 900 °C.

**Figure 4 polymers-13-00570-f004:**
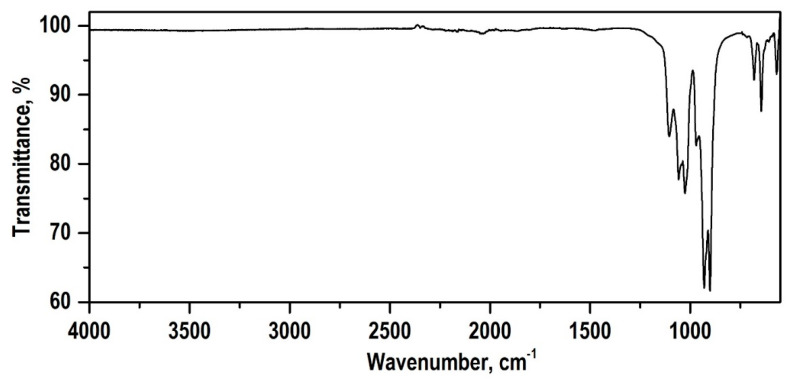
Infrared (IR) spectrum of the autoclave synthesis product sample in the CaSO_4_·2H_2_O–SiO_2_·*n*H_2_O–KOH–H_2_O system after calcination at 900 °C.

**Figure 5 polymers-13-00570-f005:**
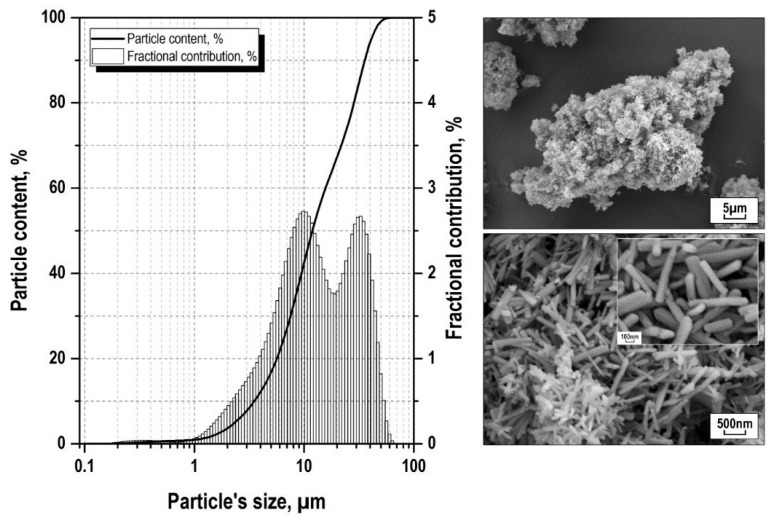
Size distribution and scanning electron microscope (SEM) image of wollastonite particles obtained after calcination at 900 °C of the autoclave synthesis product in the CaSO_4_·2H_2_O–SiO_2_·*n*H_2_O–KOH–H_2_O system.

**Figure 6 polymers-13-00570-f006:**
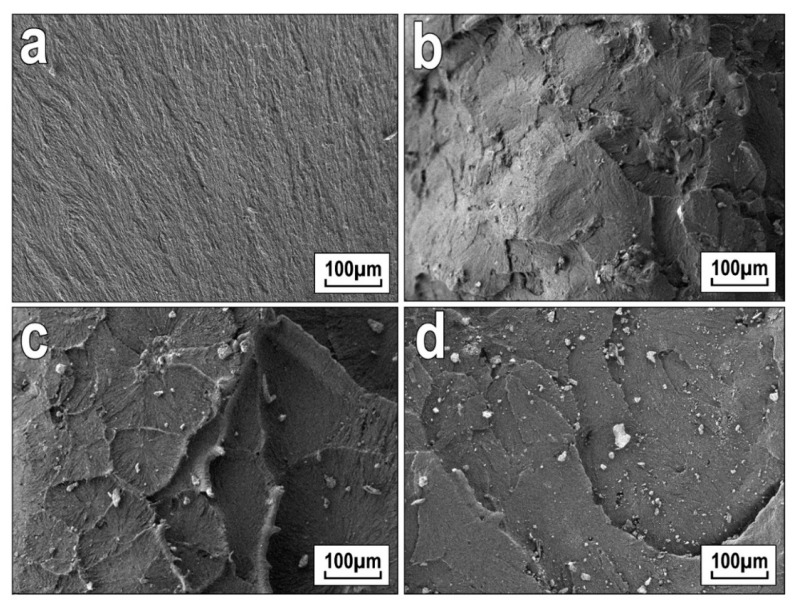
Microphotographs of the structure of (**a**) the initial ultra-high-molecular weight polyethylene (UHMWPE) and polymer composite materials (PCMs) based on UHMWPE with an addition of synthetic wollastonite (**b**) 0.5 wt.%, (**c**) 1.0 wt.%, and (**d**) 2.0 wt.%.

**Figure 7 polymers-13-00570-f007:**
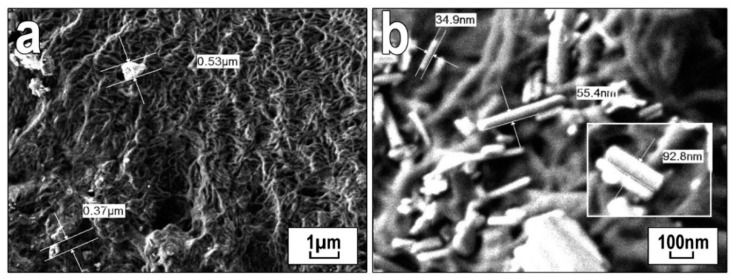
Microphotographs of supramolecular structure of PCM based on UHMWPE with an addition of 1.0 wt.% of wollastonite: (**a**) ×5000 and (**b**) ×30,000.

**Figure 8 polymers-13-00570-f008:**
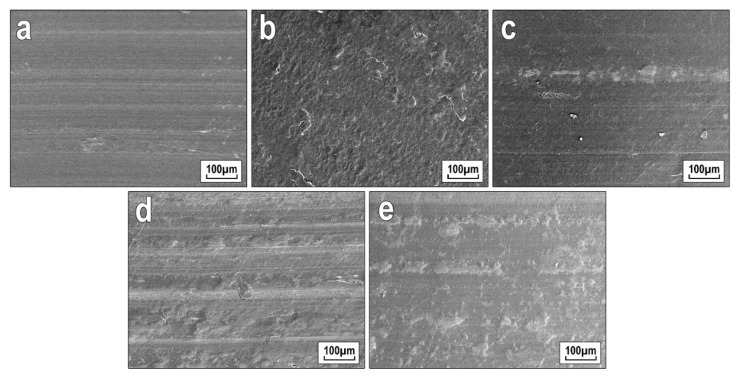
Microphotographs of the friction surfaces of PCM depending on the content of wollastonite, wt.%: (**a**) initial UHMWPE; (**b**) 1; (**c**) 2; (**d**) 10; (**e**) 20.

**Figure 9 polymers-13-00570-f009:**
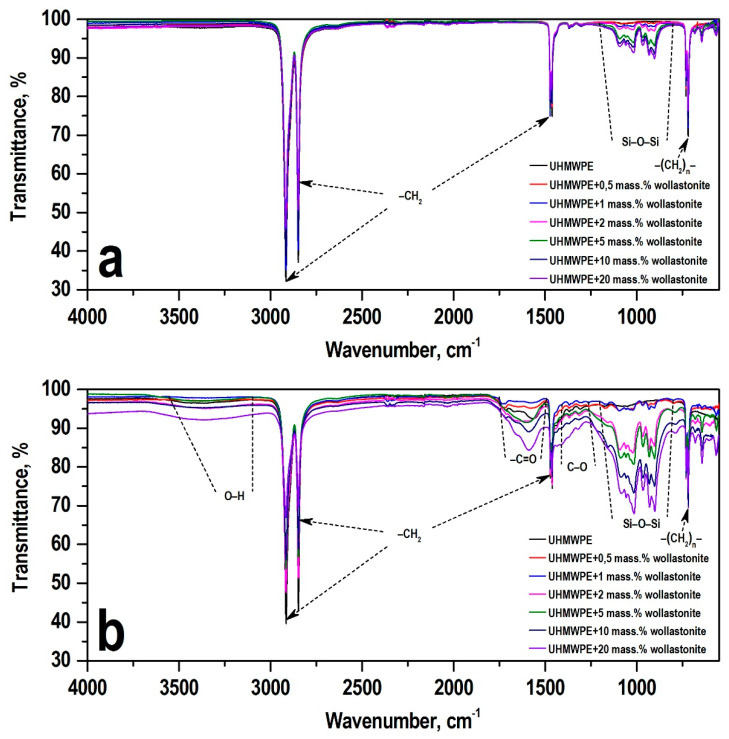
IR spectra (**a**) before friction and (**b**) after friction of PCM based on UHMWPE depending on the wollastonite content.

**Table 1 polymers-13-00570-t001:** Hardness, elongation at break, tensile strength, and Young’s modulus of ultra-high-molecular weight polyethylene (UHMWPE) and polymer composite materials (PCM).

Wollastonite Added, wt.%	Characteristics			
	Mechanical			
	H (N/mm^2^)	ε_b_, %	σ_T_, MPa	E, MPa
0	26.2 ± 1.31	311 ± 18	33 ± 2	420 ± 25
0.5	27.3 ± 1.37	329 ± 20	37 ± 1	670 ± 20
1	28.8 ± 1.44	315 ± 34	37 ± 3	574 ± 59
2	29.5 ± 1.48	355 ± 21	42 ± 3	581 ± 54
5	32.9 ± 1.64	324 ± 28	40 ± 4	630 ± 51
10	37.2 ± 1.86	308 ± 21	34 ± 2	761 ± 34
20	37.3 ± 1.87	242 ± 19	29 ± 2	698 ± 32

Notes: H—hardness, N/mm^2^; ε_b_—elongation at break, %; σ_T_—tensile strength, MPa; E—Young’s modulus, MPa.

**Table 2 polymers-13-00570-t002:** Melting point, melting enthalpy, and degree of crystallinity of PCM samples based on UHMWPE and wollastonite.

Wollastonite Added, wt.%	Characteristics		
	Thermodynamic		
	T_me_, °C	∆H_me_, J/g	α, %
0	127.7	171.1	58.8
0.5	128.1	169.0	58.4
1	127.6	167.7	58.2
2	127.7	166.1	58.2
5	127.9	162.8	58.8
10	128.0	160.9	61.4
20	128.0	151.1	64.9

Notes: T_me_—melting point, °C; ∆H_me_—melting enthalpy, J/g; α—degree of crystallinity, %.

**Table 3 polymers-13-00570-t003:** Coefficient of friction, linear wear, and wear rate of the UHMWPE and PCM.

Wollastonite Added, wt.%	Characteristics		
	Tribological		
	f	L, mm	Wear Rate, ×10^−6^ mm^3^/(N⋅m)
0	0.38 ± 0.01	0.31 ± 0.02	0.57 ± 0.01
0.5	0.40 ± 0.02	0.18 ± 0.01	0.19 ± 0.01
1	0.40 ± 0.02	0.15 ± 0.01	0.09 ± 0.01
2	0.40 ± 0.02	0.16 ± 0.02	0.37 ± 0.01
5	0.38 ± 0.01	0.12 ± 0.01	0.50 ± 0.02
10	0.36 ± 0.01	0.11 ± 0.01	0.72 ± 0.02
20	0.38 ± 0.01	0.13 ± 0.03	0.80 ± 0.01

Notes: f—coefficient of friction; L—linear wear, mm.

## Data Availability

The data presented in this study are available on request from the corresponding author.

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
