# Peer review of "UHMWPE/CaSiO3 Nanocomposite: Mechanical and Tribological Properties"

_polymers, 2021, doi:10.3390/polym13040570_

Round 1

Reviewer 1 Report

Dear author,

In this paper, the wear-resistant composites made of UHMWPE modified by needle-shaped nanosized CaSiO3 particles were studied. The effects of CaSiO3 content changing from 0 to 20 mass.% on the structure and properties of the composites were analyzed. Although nano-wollastonite filling modified polymers have been studied for several years, some positive results have been obtained in this study. The topic of the article is also consistent with the aims & scope of Polymers.

Unfortunately, there are important issues that need to be discussed and identified before the article can be recommended the publication.

  1. The keywords of this paper need to be further refined. For example, UHMWPE will not be listed separately as another keyword after the ultra-high-molecular-weight polyethylene is available, or the two are combined into one keyword.
  2. In line 111 of the paper, the composition formulation system with 0.75 mass.% of CaSiO3 in UHMWPE is listed, but the relevant content is not found in the results and discussions. What is the reason?
  3. The mechanical property indices E, εb, σb in Table 1 are changed to Ep, εp, σp in the notes below and E, eb, σb in Figure 6. Why is that?
  4. In lines 225-226 of the paper, it is mentioned that both Young's modulus and tensile strength of the composites are enhanced when the content of wollastonite is 2 mass.%. However, it is shown in Table 1 and Figure 6 that only the tensile strength is enhanced while Young's modulus is decreased. Tensile strength and elongation at break are also mentioned in the subsequent statement (lines 226-228). What are these two indicators?
  5. 3.2.2 and 3.2.3 need to be completely and carefully described and discussed. For example, what is the change rule and mechanism of each index when the addition amount of CaSiO3 changes from 0 to 20 mass.%? What is the specific amount of CaSiO3 that corresponds to the maximum and minimum change amount?
  6. In lines 238-240, it is described that with the increase of CaSiO3 content, well-dispersed particles of 1-100um size are obtained. The CaSiO3 raw material particles synthesized in the study are nanoscale, why the results are micron particles evenly dispersed in the matrix? What is the evidence for the presence of particles greater than 50um in the material?
  7. In line 243, should there be (L) after linear wear? And the unit of the following and the above figure is um, while the unit of the related index items in Table 1 is mm. Are their units different?Should there be (K) after the wear rate in lines 243-244? Should the subsequent unit mm3/N.m be mm3/ (N.m)?
  8. It is suggested that the results and their corresponding parts of the discussion should be organically integrated into one so as to avoid repeating the narration and facilitate the discussion on the formation mechanism of the results and their rules.
  9. It is stated in lines 369-371 that the friction coefficient decreases when the CaSiO3 content is 5 mass.%, and increases again when it is increased to 20 mass.%. However, as shown in Table 1 and Figure 6, it does not change. It has also been stated in 3.2.3 that the friction coefficient remains unchanged within the measurement error range (lines 254-255 and 396-397). Why is that?

  10. It is recommended to rewrite the results and conclusions to describe the results of the study in a clear and accurate one-to-one correspondence. For example, the most improvement in wear-resisting performance is increased to 6 times when the content of CaSiO3 is 1 mass.%, which is increased by 5 times.

Author Response

Response to the Reviewers’ Comments

Dear Reviewers,

We deeply appreciate the time you spent on reviewing our paper and valuable recommendations you made. All the comments are considered and corresponding changes are made to the manuscripts body text. Detailed point-by-point answers are presented below.

Reviewer #1.

In this paper, the wear-resistant composites made of UHMWPE modified by needle-shaped nanosized CaSiO3 particles were studied. The effects of CaSiO3 content changing from 0 to 20 mass.% on the structure and properties of the composites were analyzed. Although nano-wollastonite filling modified polymers have been studied for several years, some positive results have been obtained in this study. The topic of the article is also consistent with the aims & scope of Polymers.

Unfortunately, there are important issues that need to be discussed and identified before the article can be recommended the publication.

Comment #1: The keywords of this paper need to be further refined. For example, UHMWPE will not be listed separately as another keyword after the ultra-high-molecular-weight polyethylene is available, or the two are combined into one keyword.

Response to comment #1: Thank you for your comment! The keywords are combined. 

Comment #2: In line 111 of the paper, the composition formulation system with 0.75 mass.% of CaSiO3 in UHMWPE is listed, but the relevant content is not found in the results and discussions. What is the reason?

Response to comment #2: Thank you for your comment! The remark has been corrected.

Comment #3: The mechanical property indices E, εb, σb in Table 1 are changed to Ep, εp, σp in the notes below and E, eb, σb in Figure 6. Why is that?

Response to comment #3: Thank you for your comment! In the current version the symbols are as follows: εb – elongation at break, %; σT – tensile strength, MPa; Е – Young’s modulus, MPa.

Comment #4: In lines 225-226 of the paper, it is mentioned that both Young's modulus and tensile strength of the composites are enhanced when the content of wollastonite is 2 mass.%. However, it is shown in Table 1 and Figure 6 that only the tensile strength is enhanced while Young's modulus is decreased. Tensile strength and elongation at break are also mentioned in the subsequent statement (lines 226-228). What are these two indicators?

Response to comment #4: Thank you for your comment! Line 226 should have the relative elongation at break and tensile strength as shown in Table 1. Whereas the maximum increase in Young's modulus is observed at 10 wt.% filler content. The observation is corrected in the manuscript.

Comment #5: 3.2.2 and 3.2.3 need to be completely and carefully described and discussed. For example, what is the change rule and mechanism of each index when the addition amount of CaSiO3 changes from 0 to 20 mass.%? What is the specific amount of CaSiO3 that corresponds to the maximum and minimum change amount?

Response to comment #5: Thank you for your comment! Results and discussion are combined and edited. The results of changes in the deformation-strength properties of composites are explained based on structural studies. The parameters' changes of the composites' deformation-strength properties depending on the wollastonite concentration given in the paper.

Comment #6: In lines 238-240, it is described that with the increase of CaSiO3 content, well-dispersed particles of 1-100um size are obtained. The CaSiO3 raw material particles synthesized in the study are nanoscale, why the results are micron particles evenly dispersed in the matrix? What is the evidence for the presence of particles greater than 50um in the material?

Response to comment #6: Thank you for your comment! The particle size distribution analysis of wollastonite powder showed the presence of agglomerates up to 70 μm in size. The size of agglomerated wollastonite particles in the entire volume of the polymer does not reach 50 microns.

Comment #7: In line 243, should there be (L) after linear wear? And the unit of the following and the above figure is um, while the unit of the related index items in Table 1 is mm. Are their units different?

Should there be (K) after the wear rate in lines 243-244? Should the subsequent unit mm3/N.m be mm3/ (N.m)?

Response to comment #7: Thank you for your comment! Corrections made: (1) (L (mm) and (2) wear rate (k) are measured in mm3/ (N*m). 

Comment #8: It is suggested that the results and their corresponding parts of the discussion should be organically integrated into one so as to avoid repeating the narration and facilitate the discussion on the formation mechanism of the results and their rules.

Response to comment #8: Thank you for your comment! Your suggestion is taken into account. The results and discussion are combined into one section.

Comment #9: It is stated in lines 369-371 that the friction coefficient decreases when the CaSiO3 content is 5 mass.%, and increases again when it is increased to 20 mass.%. However, as shown in Table 1 and Figure 6, it does not change. It has also been stated in 3.2.3 that the friction coefficient remains unchanged within the measurement error range (lines 254-255 and 396-397). Why is that?

Response to comment #9: Thank you for your comment! The friction coefficient does not change remaining within the measurement error. Corrections are made to the manuscript.

Comment #10: It is recommended to rewrite the results and conclusions to describe the results of the study in a clear and accurate one-to-one correspondence. For example, the most improvement in wear-resisting performance is increased to 6 times when the content of CaSiO3 is 1 mass.%, which is increased by 5 times.

Response to comment #10: Thank you for your comment! The results and conclusions are rewritten.

With respect, the authors

Reviewer 2 Report

The article analyzes the effects of different proportions of CaSiO3 additives on the physical properties and friction characteristics of UHMWPE composites. The experimental design is reasonable.

The wear morphology of the composites with different proportions of additives can be clearly seen from the SEM images. The content of the experiment is very sufficient, and many kinds of research tools are used. Through the performance analysis, the characteristics of the synthesized products are studied in detail. From this aspect, I feel that the experimental research part is doing well

From the experimental data, the performance of the composite wear-resistant material is relatively excellent. It is also reasonable to explain the improvement of wear resistance and the role of CaSiO3 in different stages.

The name of the material and equipment, the manufacturer's instructions, including the operation part of the instrument, are written in detail. Characterization is well studied.

In the part of mechanism discussion, it is better to have a little theoretical depth discussion, if there is assistance in mathematical analysis and calculation, it is best.

In picture 3, the abscissa variable should be 2θ. The fonts in pictures 2 and 3 may need to be slightly adjusted. In addition, the writing format of some symbols in the article needs to be checked according to the journal.

The integration of the article is good. On the whole, I think this article is acceptable.

Author Response

Response to the Reviewers’ Comments

Dear Reviewers,

We deeply appreciate the time you spent on reviewing our paper and valuable recommendations you made. All the comments are considered and corresponding changes are made to the manuscripts body text. Detailed point-by-point answers are presented below.

Reviewer #2.

The article analyzes the effects of different proportions of CaSiO3 additives on the physical properties and friction characteristics of UHMWPE composites. The experimental design is reasonable.

The wear morphology of the composites with different proportions of additives can be clearly seen from the SEM images. The content of the experiment is very sufficient, and many kinds of research tools are used. Through the performance analysis, the characteristics of the synthesized products are studied in detail. From this aspect, I feel that the experimental research part is doing well.

Comment #1: From the experimental data, the performance of the composite wear-resistant material is relatively excellent. It is also reasonable to explain the improvement of wear resistance and the role of CaSiO3 in different stages.

Response to comment #1: Thank you for your comment! The discussion now presents an explanation of the effect of wollastonite particles on the wear resistance of UHMWPE and polymer composite material in page 18. The name of the material and equipment, the manufacturer's instructions, including the operation part of the instrument, are written in detail.

Comment #2: In the part of mechanism discussion, it is better to have a little theoretical depth discussion, if there is assistance in mathematical analysis and calculation, it is best.

Response to comment #2: Thank you for your comment! The results were additionally discussed. Statistical processing of the experimental data was performed using standard methods of mathematical statistics, determining the values of the sample mean square deviation and the limits of confidence intervals by Student's test at a reliability level of 0.95. The number of tests is given in the research methods section.

Comment #3: In picture 3, the abscissa variable should be 2θ. The fonts in pictures 2 and 3 may need to be slightly adjusted. In addition, the writing format of some symbols in the article needs to be checked according to the journal. The integration of the article is good. On the whole, I think this article is acceptable.

Response to comment #3: Thank you for your comment! Unfortunately, it is not entirely clear what Reviewer means when asks “The fonts in pictures 2 and 3 may need to be slightly adjusted“. We are glad to make any corrections to the graphics in our paper to make the design better, more understandable, more visually pleasing to the reader. But, unfortunately, in this case we do not quite understand what is required. In our opinion, the design of Figures 2 and 3 does not need to be corrected.

With respect, the authors

Reviewer 3 Report

Did Authors not melt the polymer during compounding? Because it seems strange. How then the Authors abtained good dispersion of filler?

"When filling UHMWPE with 2 mass.% of wollastonite, in PCM there is an increase in the value of Young’s modulus and tensile strength (Fig. 6)."

Actually Figure 6, as well as Table 1 indicates otherwise.

Why the values of modulus are so inconsistent with the filler loading?

If tribooxiadation is noted during wear tests I recommend to compare thermal stability of composites before and after testing.

Author Response

Response to the Reviewers’ Comments

Dear Reviewers,

We deeply appreciate the time you spent on reviewing our paper and valuable recommendations you made. All the comments are considered and corresponding changes are made to the manuscripts body text. Detailed point-by-point answers are presented below.

Reviewer #3.

Comment #1: Did Authors not melt the polymer during compounding? Because it seems strange. How then the Authors abtained good dispersion of filler?

Response to comment #1: Thank you for your comment! Samples were obtained using hot pressing after mixing the polymer mixture in a paddle mixer. Thus, a dry composite mixture with a filler was obtained first which was further processed by hot pressing.

Comment #2: "When filling UHMWPE with 2 mass.% of wollastonite, in PCM there is an increase in the value of Young’s modulus and tensile strength (Fig. 6)."  Actually Figure 6, as well as Table 1 indicates otherwise.

Response to comment #2: Thank you for your comment! The text was referring to an increase in elongation at break as well as in tensile strength.

Comment #3: Why the values of modulus are so inconsistent with the filler loading?

Response to comment #3: Thank you for your comment! The addition of wollastonite filler to UHMWPE leads to an increase in the value of Young's modulus of the formed polymer composite. As the results of our study show, the filler concentration is 10 wt.% concentration of the filler increases Young's modulus by 81 %. While in PCM containing 20 wt.% increases to 66 %. The decrease in the value of Young's modulus when the concentration of wollastonite filler increases to 20 wt.% is explained by the change in the deformation characteristics as a result of the formation of defective areas in the composite material. In Table 1 the composite containing 20 wt.% wollastonite has the lowest values of the relative elongation at break and tensile strength.

Comment #4: If tribooxiadation is noted during wear tests I recommend to compare thermal stability of composites before and after testing.

Response to comment #4: Thank you for the very constructive comment! In the future, we will use the suggested recommendation while the study the effect of temperature increase on the final properties of composites (including the expansion of thermal stability methods). However, there are already data have been presented are indicating that at the similar wear modes which being used in our work the temperature in the friction zone does not exceed 45 °C [10.3103/S1068366620020129]. This temperature is much lower than the heat resistance temperature of UHMWPE (229 °C [10.1155/2019/8687450]). Our earlier TGA studies showed that the thermostability of UHMWPE-based samples does not depend on the degree of filling. These data are confirmed by calculations of the activation energy by the Freeman-Carrol method. The activation energy remains unchanged, and the thermostability values do not change in comparison with the UHMWPE and remain within the selected matrix-filler ratios.

With respect, the authors

Round 2

Reviewer 1 Report

  1. In Table 1 of the new draft, the E value changed from 588 of the original to 420 when the content of wollastonite was wt.1%. Please give the reason and basis for the change in the data.
  2. The crystallinity data in Table 2 of the new draft are completely different from the original data. Please give reasons and basis for the change in these data.
  3. Is 26% correct in line 444? Please also check the other data carefully and express it accurately!

Author Response

Response to the Reviewers’ Comments

Dear Reviewer,

We deeply appreciate the time you spent on reviewing our paper and valuable recommendations you made. All the comments are considered and corresponding changes are made to the manuscripts body text. Detailed point-by-point answers are presented below.

Comment #1: In Table 1 of the new draft, the E value changed from 588 of the original to 420 when the content of wollastonite was wt.1%. Please give the reason and basis for the change in the data.

Response to comment #1:  Thank you for your important comment. The E value of the original UHMWPE has been changed in Table 1 because the old version used data calculated not according to ASTM standard. The E value of composites containing wollastonite is correct.

Comment #2: The crystallinity data in Table 2 of the new draft are completely different from the original data. Please give reasons and basis for the change in these data.

Response to comment #2: Thank you for your comment. The PCM crystallinity data in Table 1 was calculated using Formula 2 taking into account the filler content.

Comment #3: Is 26% correct in line 444? Please also check the other data carefully and express it accurately!

Response to comment #3: Thank you for your comment. Yes, there was an error. The hardness of the composite increases by 42 %.
